# Policy Continuation with Hindsight Inverse Dynamics

**Hao Sun[1], Zhizhong Li[1], Xiaotong Liu[2], Dahua Lin[1], Bolei Zhou[1]**
[1]The Chinese University of Hong Kong, [2]Peking University

## Abstract

Solving goal-oriented tasks is an important but challenging problem in reinforcement learning (RL). For such tasks, the rewards are often sparse, making it difficult to learn a policy effectively. To tackle this difficulty, we propose a new approach called *Policy Continuation with Hindsight Inverse Dynamics (PCHID)*. This approach learns from Hindsight Inverse Dynamics based on Hindsight Experience Replay. Enabling the learning process in a self-imitated manner and thus can be trained with supervised learning. This work also extends it to multi-step settings with Policy Continuation. The proposed method is general, which can work in isolation or be combined with other on-policy and off-policy algorithms. On two multi-goal tasks GridWorld and FetchReach, PCHID significantly improves the sample efficiency as well as the final performance[1].

## 1 Introduction

Imagine you are given the task of Tower of Hanoi with ten disks, what would you probably do to solve this complex problem? This game looks daunting at the first glance. However through trials and errors, one may discover the key is to recursively relocate the disks on the top of the stack from one pod to another, assisted by an intermediate one. In this case, you are actually learning skills from easier sub-tasks and those skills help you to learn more. This case exemplifies the procedure of self-imitated curriculum learning, which recursively develops the skills of solving more complex problems.

Tower of Hanoi belongs to an important kind of challenging problems in Reinforcement Learning (RL), namely solving the goal-oriented tasks. In such tasks, rewards are usually very sparse. For example, in many goal-oriented tasks, a single binary reward is provided only when the task is completed [1, 2, 3]. Previous works attribute the difficulty of the sparse reward problems to the low efficiency in experience collection [4]. Thus many approaches have been proposed to tackle this problem, including automatic goal generation [5], self-imitation learning [6], hierarchical reinforcement learning [7], curiosity driven methods [8, 9], curriculum learning [1, 10], and Hindsight Experience Replay (HER) [11]. Most of these works guide the agent by demonstrating on successful choices based on sufficient exploration to improve learning efficiency. Differently, HER opens up a new way to learn more from failures, assigning hindsight credit to primal experiences. However, it is limited by only applicable when combined with off-policy algorithms[3].

In this paper we propose an approach of goal-oriented RL called Policy Continuation with Hindsight Inverse Dynamics (PCHID), which leverages the key idea of self-imitate learning. In contrast to HER, our method can work as an auxiliary module for both on-policy and off-policy algorithms, or as an isolated controller itself. Moreover, by learning to predict actions directly from back-propagation through self-imitation [12], instead of temporal difference [13] or policy gradient [14, 15, 16, 17], the data efficiency is greatly improved.

The contributions of this work lie in three aspects: **(1)** We introduce the state-goal space partition for multi-goal RL and thereon define Policy Continuation (PC) as a new approach to such tasks.

**(2)** We propose Hindsight Inverse Dynamics (HID), which extends the vanilla Inverse Dynamics method to the goal-oriented setting. **(3)** We further integrate PC and HID into PCHID, which can effectively leverage self-supervised learning to accelerate the process of reinforcement learning. Note that PCHID is a general method. Both on-policy and off-policy algorithms can benefit therefrom. We test this method on challenging RL problems, where it achieves considerably higher sample efficiency and performance.

## 2    Related Work

**Hindsight Experience Replay**    Learning with sparse rewards in RL problems is always a leading challenge for the rewards are usually uneasy to reach with random explorations. Hindsight Experience Replay (HER) which relabels the failed rollouts as successful ones is proposed by Andrychowicz et al. [11] as a method to deal with such problem. The agent in HER receives a reward when reaching either the original goal or the relabeled goal in each episode by storing both original transition pairs $s_t, g, a_t, r$ and relabeled transitions $s_t, g', a_t, r'$ in the replay buffer. HER was later extended to work with demonstration data [4] and boosted with multi-processing training [3]. The work of Rauber et al. [18] further extended the hindsight knowledge into policy gradient methods using importance sampling.

**Inverse Dynamics**    Given a state transition pair $(s_t, a_t, s_{t+1})$, the inverse dynamics  [19] takes $(s_t, s_{t+1})$ as the input and outputs the corresponding action $a_t$. Previous works used inverse dynamics to perform feature extraction [20, 9, 21] for policy network optimization. The actions stored in such transition pairs are always collected with a random policy so that it can barely be used to optimize the policy network directly. In our work, we use hindsight experience to revise the original transition pairs in inverse dynamics, and we call this approach Hindsight Inverse Dynamics. The details will be elucidated in the next section.

**Auxiliary Task and Curiosity Driven Method**    Mirowski et al. [22] propose to jointly learn the goal-driven reinforcement learning problems with an unsupervised depth prediction task and a self-supervised loop closure classification task, achieving data efficiency and task performance improvement. But their method requires extra supervision like depth input. Shelhamer et al. [21] introduce several self-supervised auxiliary tasks to perform feature extraction and adopt the learned features to reinforcement learning, improving the data efficiency and returns of end-to-end learning. Pathak et al. [20] propose to learn an intrinsic curiosity reward besides the normal extrinsic reward, formulated by prediction error of a visual feature space and improved the learning efficiency. Both of the approaches belong to self-supervision and utilize inverse dynamics during training. Although our method can be used as an auxiliary task and trained in self-supervised way, we improve the vanilla inverse dynamics with hindsight, which enables direct joint training of policy networks with temporal difference and self-supervised learning.

## 3    Policy Continuation with Hindsight Inverse Dynamics

In this section we first briefly go through the preliminaries in Sec.3.1. In Sec.3.2 we retrospect a toy example introduced in HER as a motivating example. Sec.3.3 to 3.6 describe our method in detail.

### 3.1    Preliminaries

**Markov Decision Process**    We consider a Markov Decision Process (MDP) denoted by a tuple $(\mathcal{S}, \mathcal{A}, \mathcal{P}, r, \gamma)$, where $\mathcal{S}$, $\mathcal{A}$ are the finite state and action space, $\mathcal{P}$ describes the transition probability as $\mathcal{S} \times \mathcal{A} \times \mathcal{S} \rightarrow [0, 1]$. $r : \mathcal{S} \rightarrow \mathbb{R}$ is the reward function and $\gamma \in [0, 1]$ is the discount factor. $\pi : \mathcal{S} \times \mathcal{A} \rightarrow [0, 1]$ denotes a policy, and an optimal policy $\pi^*$ satisfies $\pi^* = \arg\max_\pi \mathbb{E}_{s,a\sim\pi}[\sum_{t=0}^{\infty} \gamma^t r(s_t)]$ where $a_t \sim \pi(a_t|s_t)$, $s_{t+1} \sim \mathcal{P}(s_{t+1}|a_t, s_t)$ and an $s_0$ is given as a start state. When transition and policy are deterministic, $\pi^* = \arg\max_\pi \mathbb{E}_{s_0}[\sum_{t=0}^{\infty} \gamma^t r(s_t)]$ and $a_t = \pi(s_t)$, $s_{t+1} = \mathcal{T}(s_t, a_t)$, where $\pi : \mathcal{S} \rightarrow \mathcal{A}$ is deterministic and $\mathcal{T}$ models the deterministic transition dynamics. The expectation is over all the possible start states.

**Universal Value Function Approximators and Multi-Goal RL**    The *Universal Value Function Approximator (UVFA)* [23] extends the state space of *Deep Q-Networks (DQN)* [24] to include goal

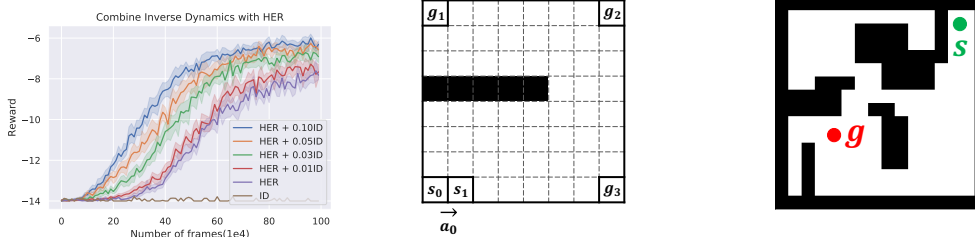

Figure 1: (a): Results in bit-flipping problem. (b): An illustration of flat state space. (c): An example of the GridWorld domain, which is a non-flat case.

state $g \in \mathcal{G}$ as part of the input, $i.e.$, $s_t$ is extended to $(s_t, g) \in \mathcal{S} \times \mathcal{G}$. And the policy becomes $\pi : \mathcal{S} \times \mathcal{G} \to a_t$, which is pretty useful in the setting where there are multiple goals to achieve. Moreover, Schaul $et\,al.$ [23] show that in such a setting, the learned policy can be generalized to previous unseen state-goal pairs. Our application of UVFA on Proximal Policy Optimization algorithm (PPO) [25] is straightforward. In the following of this work, we will use state-goal pairs to denote the *extended state space* $(s, g) \in \mathcal{S} \times \mathcal{G}$, $a_t = \pi(s_t, g)$ and $(s_{t+1}, g) = \mathcal{T}(s_t, a_t)$. The goal $g$ is fixed within an episode, but changed across different episodes.

## 3.2 Revisiting the Bit-Flipping Problem

The bit-flipping problem was provided as a motivating example in HER [11], where there are $n$ bits with the state space $\mathcal{S} = \{0, 1\}^n$ and the action space $\mathcal{A} = \{0, 1, ..0, n-1\}$. An action $a$ corresponds to turn the $a$-th bit of the state. Each episode starts with a randomly generated state $s_0$ and a random goal state $g$. Only when the goal state $g$ is reached the agent will receive a reward. HER proposed to relabel the failed trajectories to receive more reward signals thus enable the policy to learn from failures. However, the method is based on temporal difference thus the efficiency of data is limited. As we can learn from failures, here comes the question that can we learn a policy by supervised learning where the data is generated using hindsight experience?

Inspired by the self-imitate learning ability of human, we aim to employ self-imitation to learn how to get success in RL even when the original goal has not yet achieved. A straightforward way to utilize self-imitate learning is to adopt the inverse dynamics. However, in most cases the actions stored in inverse dynamics are irrelevant to the goals.

Specifically, transition tuples like $((s_t, g), (s_{t+1}, g), a_t)$ are saved to learn the inverse dynamics of goal-oriented tasks. Then the learning process can be executed simply as classification when action space is discrete or regression when action space is continuous. Given a neural network parameterized by $\phi$, the objective of learning inverse dynamics is as follows,

$$\phi = \arg\min_{\phi} \sum_{s_t, s_{t+1}, a_t} ||f_{\phi}((s_t, g), (s_{t+1}, g)) - a_t||^2. \tag{1}$$

Due to the unawareness of the goals while the agent is taking actions, the goals $g$ in Eq.(1) are only placeholders. Thus, it will cost nothing to replace $g$ with $g' = m(s_{t+1})$ but result in a more meaningful form, $i.e.$, encoding the following state as a hindsight goal. That is to say, if the agent wants to reach $g'$ from $s_t$, it should take the action of $a_t$, thus the decision making process is aware of the hindsight goal. We adopt $f_{\phi}$ trained from Eq.(1) as an additional module incorporating with HER in the Bit-flipping environment, by simply adding up their logit outputs. As shown in Fig.1(a), such an additional module leads to significant improvement. We attribute this success to the flatness of the state space. Fig.1(b) illustrates such a flatness case where an agent in a grid map is required to reach the goal $g_3$ starting from $s_0$: if the agent has already known how to reach $s_1$ in the east, intuitively, it has no problem to extrapolate its policy to reach $g_3$ in the farther east.

Nevertheless, success is not always within an effortless single step reach. Reaching the goals of $g_1$ and $g_2$ are relatively harder tasks, and navigating from the start point to goal point in the GridWorld domain shown in Fig.1(c) is even more challenging. To further employ the self-imitate learning and overcome the single step limitation of inverse dynamics, we come up with a new approach called Policy Continuation with Hindsight Inverse Dynamics.

### 3.3 Perspective of Policy Continuation on Multi-Goal RL Task

Our approach is mainly based on policy continuation over sub-policies, which can be viewed as an emendation of the spontaneous extrapolation in the bit-flipping case.

**Definition 1: Policy Continuation(PC)** *Suppose $\pi$ is a policy function defined on a non-empty sub-state-space $\mathcal{S}_U$ of the state space $\mathcal{S}$, i.e., $\mathcal{S}_U \subset \mathcal{S}$. If $\mathcal{S}_V$ is a larger subset of $\mathcal{S}$, containing $\mathcal{S}_U$, i.e., $\mathcal{S}_U \subset \mathcal{S}_V$ and $\Pi$ is a policy function defined on $\mathcal{S}_V$ such that*

$$\Pi(s) = \pi(s) \qquad \forall s \in \mathcal{S}_U$$

*then we call $\Pi$ a policy continuation of $\pi$, or we can say the restriction of $\Pi$ to $\mathcal{S}_U$ is the policy function $\pi$.*

Denote the optimal policy as $\pi^* : (s_t, g_t) \to a_t$, we introduce the concept of $k$-step solvability:

**Definition 2: $k$-Step Solvability** *Given a state-goal pair $(s, g)$ as a task of a certain system with deterministic dynamics, if reaching the goal $g$ needs at least $k$ steps under the optimal policy $\pi^*$ starting from $s$, i.e., starting from $s_0 = s$ and execute $a_i = \pi^*(s_i, g)$ for $i = \{0, 1, ..., k-1\}$, the state $s_k = \mathcal{T}(s_{k-1}, a_{k-1})$ satisfies $m(s_k) = g$, we call the pair $(s, g)$ has $k$-step solvability, or $(s, g)$ is $k$-step solvable.*

Ideally the k-step solvability means the number of steps it should take from $s$ to $g$, given the maximum permitted action value. In practice the $k$-step solvability is an evolving concept that can gradually change during the learning process, thus is defined as "whether it can be solve with $\pi_{k-1}$ within $k$ steps after the convergence of $\pi_{k-1}$ trained on $(k$-1$)$-step HIDs".

We follow HER to assume a mapping $m : \mathcal{S} \to \mathcal{G}$ s.t. $\forall s \in \mathcal{S}$ the reward function $r(s, m(s)) = 1$, thus, the information of a goal $g$ is encoded in state $s$. For the simplest case we have $m$ as identical mapping and $\mathcal{G} = \mathcal{S}$ where the goal $g$ is considered as a certain state $s$ of the system.

Following the idea of recursion in curriculum learning, we can divide the finite state-goal space into $T + 2$ parts according to their $k$-step solvability,

$$\mathcal{S} \times \mathcal{G} = (\mathcal{S} \times \mathcal{G})_0 \cup (\mathcal{S} \times \mathcal{G})_1 \cup ... \cup (\mathcal{S} \times \mathcal{G})_T \cup (\mathcal{S} \times \mathcal{G})_U \tag{2}$$

where $(s, g) \in \mathcal{S} \times \mathcal{G}$, $T$ is a finite time-step horizon that we suppose the task should be solved within, and $(\mathcal{S} \times \mathcal{G})_i, i \in \{0, 1, 2, ...T\}$ denotes the set of $i$-step solvable state-goal pairs, $(s, g) \in (\mathcal{S} \times \mathcal{G})_U$ denotes unsolvable state-goal pairs, *i.e.*, $(s, g)$ is not $k$-step solvable for $\forall k \in \{0, 1, 2, ..., T\}$, and $(\mathcal{S} \times \mathcal{G})_0$ is the trivial case $g = m(s_0)$. As the optimal policy only aims to solve the solvable state-goal pairs, we can take $(\mathcal{S} \times \mathcal{G})_U$ out of consideration. It is clear that we can define a disjoint sub-state-goal space union for the solvable state-goal pairs

**Definition 3: Solvable State-Goal Space Partition** *Given a certain environment, any solvable state-goal pairs can be categorized into only one sub state-goal space by the following partition*

$$\mathcal{S} \times \mathcal{G} \backslash (\mathcal{S} \times \mathcal{G})_U = \bigcup_{j=0}^{T} (\mathcal{S} \times \mathcal{G})_j \tag{3}$$

Then, we define a set of sub-policies $\{\pi_i\}, i \in \{0, 1, 2, ..., T\}$ on solvable sub-state-goal space $\bigcup_{j=0}^{i} (\mathcal{S} \times \mathcal{G})_j$ respectively, with the following definition

**Definition 4: Sub Policy on Sub Space** *$\pi_i$ is a sub-policy defined on the sub-state-goal space $(\mathcal{S} \times \mathcal{G})_i$. We say $\pi_i^*$ is an optimal sub-policy if it is able to solve all $i$-step solvable state-goal pair tasks in $i$ steps.*

**Corollary 1:** *If $\{\pi_i^*\}$ is restricted as a policy continuation of $\{\pi_{i-1}^*\}$ for $\forall i \in \{1, 2, ...k\}$, $\pi_i^*$ is able to solve any $i$-step solvable problem for $i \leq k$. By definition, the optimal policy $\pi^*$ is a policy continuation of the sub policy $\pi_T^*$, and $\pi_T^*$ is already a substitute for the optimal policy $\pi^*$.*

We can recursively approximate $\pi^*$ by expanding the domain of sub-state-goal space in policy continuation from an optimal sub-policy $\pi_0^*$. While in practice, we use neural networks to approximate

such sub-policies to do policy continuation. We propose to parameterize a policy function $\pi = f_\theta$ by $\theta$ with neural networks and optimize $f_\theta$ by self-supervised learning with the data collected by Hindsight Inverse Dynamics (HID) recursively and optimize $\pi_i$ by joint optimization.

## 3.4 Hindsight Inverse Dynamics

**One-Step Hindsight Inverse Dynamics** One step HID data can be collected easily. With $n$ randomly rollout trajectories $\{(s_0, g), a_0, r_0, (s_1, g), a_1, ..., (s_T, g), a_T, r_T\}_i, i \in \{1, 2, ..., n\}$, we can use a modified inverse dynamics by substituting the original goal $g$ with hindsight goal $g' = m(s_{t+1})$ for every $s_t$ and result in $\{(s_0, m(s_1)), a_0, (s_1, m(s_2)), a_1, ..., (s_{T-1}, m(s_T)), a_{T-1}\}_i, i \in \{1, 2, ..., n\}$. We can then fit $f_{\theta_1}$ by

$$\theta_1 = \arg\min_\theta \sum_{s_t, s_{t+1}, a_t} ||f_\theta((s_t, m(s_{t+1})), (s_{t+1}, m(s_{t+1}))) - a_t||^2 \tag{4}$$

By collecting enough trajectories, we can optimize $f_\theta$ implemented by neural networks with stochastic gradient descent [26]. When $m$ is an identical mapping, the function $f_{\theta_1}$ is a good enough approximator for $\pi_1^*$, which is guaranteed by the approximation ability of neural networks [27, 28, 29]. Otherwise, we should adapt Eq. 4 as $\theta_1 = \arg\min_\theta \sum_{s_t, s_{t+1}, a_t} ||f_\theta((s_t, m(s_{t+1})), m(s_{t+1})) - a_t||^2$, *i.e.*, we should omit the state information in future state $s_{t+1}$, to regard $f_{\theta_1}$ as a policy. And in practice it becomes $\theta_1 = \arg\min_\theta \sum_{s_t, s_{t+1}, a_t} ||f_\theta(s_t, m(s_{t+1})) - a_t||^2$.

**Multi-Step Hindsight Inverse Dynamics** Once we have $f_{\theta_{k-1}}$, an approximator of $\pi_{k-1}^*$, $k$-step HID is ready to get. We can collect valid $k$-step HID data recursively by testing whether the $k$-step HID state-goal pairs indeed need $k$ steps to solve, i.e., for any $k$-step transitions $\{(s_t, g), a_t, r_t, ..., (s_{t+k}, g), a_{t+k}, r_{t+k}\}$, if our policy $\pi_{k-1}^*$ at hand can not provide with another solution from $(s_t, m(s_{t+k}))$ to $(s_{t+k}, m(s_{t+k}))$ in less than $k$ steps, the state-goal pair $(s_t, m(s_{t+k}))$ must be $k$-step solvable, and this pair together with the action $a_t$ will be marked as $(s_t^{(k)}, m(s_{t+k}^{(k)})), a_t^{(k)}$. Fig.2 illustrates this process. The testing process is based on a function $\text{TEST}(\cdot)$ and we will focus on the selection of TEST in Sec.3.6. Transition pairs like this will be collected to optimize $\theta_k$. In practice, we leverage joint training to ensure $f_{\theta_k}$ to be a policy continuation of $\pi_i^*, i \in \{1, ..., k\}$ i.e.,

$$\theta_k = \arg\min_\theta \sum_{s_t^{(i)}, s_{t+i}^{(i)}, a_t^{(i)}, i \in \{1, ..., k\}} ||f_\theta((s_t, m(s_{t+i})), (s_{t+i}, m(s_{t+i}))) - a_t||^2 \tag{5}$$

## 3.5 Dynamic Programming Formulation

For most goal-oriented tasks, the learning objective is to find a policy to reach the goal as soon as possible. In such circumstances,

$$L^\pi(s_t, g) = L^\pi(s_{t+1}, g) + 1 \tag{6}$$

where $L^\pi(s, g)$ here is defined as the number of steps to be executed from $s$ to $g$ with policy $\pi$ and 1 is the additional

$$L^{\pi^*}(s_t, g) = L^{\pi^*}(s_{t+1}, g) + 1 \tag{7}$$

and $\pi^* = \arg\min_\pi L^\pi(s_t, g)$ As for the learning process, it is impossible to enumerate all possible intermediate state $s_{t+1}$ in the continuous state space and

Suppose now we have the optimal sub-policy $\pi_{k-1}^*$ of all $i$-step solvable problems $\forall i \leq k-1$, we will have

$$L^{\pi_k^*}(s_t, g) = L^{\pi_{k-1}^*}(s_{t+1}, g) + 1 \tag{8}$$

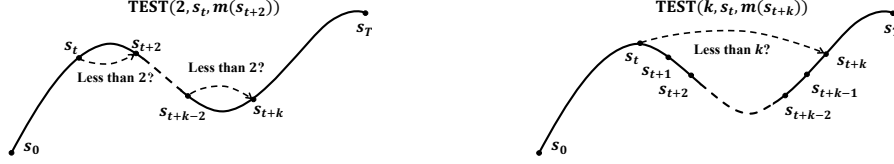

Figure 2: Test whether the transitions are 2-step (left) or $k$-step (right) solvable. The TEST function returns True if the transition $s_t \rightarrow s_{t+k}$ needs at least $k$ steps.

---

**Algorithm 1** Policy Continuation with Hindsight Inverse Dynamics (PCHID)

---

  **Require** policy $\pi_b(s, g)$, reward function $r(s, g)$ (equal to 1 if $g = m(s)$ else 0), a buffer for PCHID $\mathbb{B} = \{\mathbb{B}_1, \mathbb{B}_2, ..., \mathbb{B}_{T-1}\}$, a list $\mathbb{K}$
  Initialize $\pi_b(s, g)$, $\mathbb{B}$, $\mathbb{K} = [1]$
  **for** episode $= 1, M$ **do**
     generate $s_0, g$ by the system
     **for** $t = 0, T - 1$ **do**
        Select an action by the behavior policy $a_t = \pi_b(s_t, g)$
        Execute the action $a_t$ and get the next state $s_{t+1}$
        Store the transition $((s_t, g), a_t, (s_{t+1}, g))$ in a temporary episode buffer
     **end for**
     **for** $t = 0, T - 1$ **do**
        **for** $k \in \mathbb{K}$ **do**
           calculate additional goal according to $s_{t+k}$ by $g' = m(s_{t+k})$
           **if** TEST($k, s_t, g'$) = True **then**
              Store $(s_t, g', a_t)$ in $\mathbb{B}_k$
           **end if**
        **end for**
     **end for**
     Sample a minibatch $B$ from buffer $\mathbb{B}$
     Optimize behavior policy $\pi_b(s_t, g')$ to predict $a_t$ by supervised learning
     **if** Converge **then**
        Add $\max(\mathbb{K}) + 1$ in $\mathbb{K}$
     **end if**
  **end for**

---

holds for any $(s_t, g) \in (\mathcal{S} \times \mathcal{G})_k$. We can sample trajectories by random rollout or any unbiased policies and choose some feasible $(s_t, g)$ pairs from them, $i.e.$, any $s_t$ and $s_{t+k}$ in a trajectory that can not be solved by the

$$s_t \xrightarrow{a_t = \pi_k(s_t, s_{t+k})} s_{t+1} \xrightarrow{\pi_{k-1}^*} s_{t+k} \qquad (9)$$

Such a recursive approach starts from $\pi_1^*$, which can be easily approximated by trained with self supervised learning by any given $(s_t, s_{t+1})$ pairs for $(s_t, s_{t+1}) \in (\mathcal{S} \times \mathcal{G})_1$ by definition.

The combination of PC and with multi-step HID leads to our algorithm PCHID. PCHID can work alone or as an auxiliary module with other RL algorithms. We discuss three different combination methods of PCHID and other algorithms in Sec.4.3. The full algorithm of the PCHID is presented as Algorithm 1.

### 3.6 On the Selection of TEST Function

In Algorithm 1, a crucial step to extend the $(k - 1)$-step sub policy to $k$-step sub policy is to test whether a $k$-step transition $s_t \rightarrow s_{t+k}$ in a trajectory is indeed a $k$-step solvable problem if we regard $s_t$ as a start state $s_0$ and $m(s_{t+k})$ as a goal $g$. We propose two approaches and evaluate both in Sec.4.

**Interaction**  A straightforward idea is to reset the environment to $s_t$ and execute action $a_t$ by policy $\pi_{k-1}$, followed by execution of $a_{t+1}, a_{t+2}, ...$, and record if it achieves the goal in less than $k$ steps.

We call this approach *Interaction* for it requires the environment to be resettable and interact with the environment. This approach can be portable when the transition dynamics is known or can be approximated without a heavy computation expense.

**Random Network Distillation (RND)**    Given a state as input, the RND [30] is proposed to provide exploration bonus by comparing the output difference between a fixed randomly initialized neural network $N_A$ and another neural network $N_B$, which is trained to minimize the output difference between $N_A$ and $N_B$ with previous states. After training $N_B$ with $1, 2, ..., k-1$ step transition pairs to minimize the output difference between $N_A$ and $N_B$, since $N_B$ has never seen $k$-step solvable transition pairs, these pairs will be differentiated for they lead to larger output differences.

### 3.7  Synchronous Improvement

In PCHID, the learning scheme is set to be curriculum, *i.e.*, the agent must learn to master easy skills before learning complex ones. However, in general the efficiency of finding a transition sequence that is $i$-step solvable decreases as $i$ increases. The size of buffer $\mathbb{B}_i$ is thus decreasing for $i = 1, 2, 3, ..., T$ and the learning of $\pi_i$ might be restricted due to limited experiences. Besides, in continuous control tasks, the $k$-step solvability means the number of steps it should take from $s$ to $g$, given the maximum permitted action value. In practice the $k$-step solvability can be treated as an evolving concept that changes gradually as the learning goes. Specifically, at the beginning, an agent can only walk with small paces as it has learned from experiences collected by random movements. As the training continues, the agent is confident to move with larger paces, which may change the distribution of selected actions. Consequently, previous $k$-step solvable state goal pairs may be solved in less than $k$ steps.

Based on the efficiency limitation and the progressive definition of $k$-step solvatbility, we propose a synchronous version of PCHID. Readers please refer to the supplementary material for detailed discussion on the intuitive interpretation and empirical results.

## 4  Experiments

As a policy $\pi(s, g)$ aims at reaching a state $s'$ where $m(s') = g$, by intuition the difficulty of solving such a goal-oriented task depends on the complexity of $m$. In Sec.4.1 we start with a simple case where $m$ is an identical mapping in the environment of GridWorld by showing the agent a fully observable map. Moreover, the GridWorld environment permits us to use prior knowledge to calculate the accuracy of any TEST function. We show that PCHID can work independently or augmented with the DQN in discrete action space setting, outperforming the DQN as well as the DQN augmented with HER. The GridWorld environment corresponds to the identical mapping case $\mathcal{G} = \mathcal{S}$. In Sec.4.2 we test our method on a continuous control problem, the FetchReach environment provided by Plappert et al. [3]. Our method outperforms PPO by achieving $100\%$ successful rate in about $100$ episodes. We further compare the sensitivity of PPO to reward values and the robustness PCHID owns. The state-goal mapping of FetchReach environment is $\mathcal{G} \subset \mathcal{S}$.

### 4.1  GridWorld Navigation

We use the GridWorld navigation task in Value Iteration Networks (VIN) [31], in which the state information includes the position of the agent, and an image of the map of obstacles and goal position. In our experiments we use $16 \times 16$ domains, navigation in which is not an effortless task. Fig.1(c) shows an example of our domains. The action space is discrete and contains $8$ actions leading the agent to its $8$ neighbour positions respectively. A reward of $10$ will be provided if the agent reaches the goal within $50$ timesteps, otherwise the agent will receive a reward of $-0.02$. An action leading the agent to an obstacle will not be executed, thus the agent will stay where it is. In each episode, a new map will randomly selected start $s$ and goal $g$ points will be generated. We train our agent for $500$ episodes in total so that the agent needs to learn to navigate within just $500$ trials, which is much less than the number used in VIN [31].[2] Thus we can demonstrate the high data efficiency of PCHID

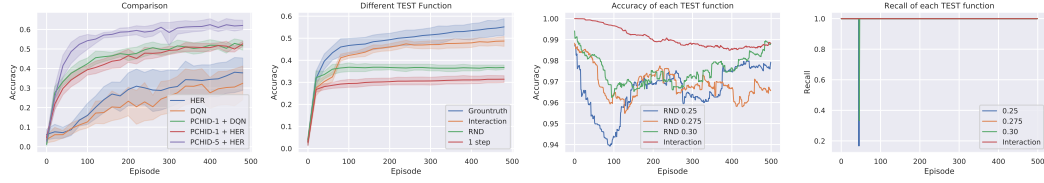

Figure 3: (a): The rollout success rate on test maps in 10 experiments with different random seeds. HER outperforms VIN, but the difference disappears when combined with PCHID. PCHID-1 and PCHID-5 represent 1-step and 5-step PCHID. (b): Performance of PCHID module alone with different TEST functions. The blue line is from ground truth testing results, the orange line and green line are Interaction and RND respectively, and the red line is the 1-step result as a baseline. (c)(d): Test accuracy and recall with Interaction and RND method under different threshold.

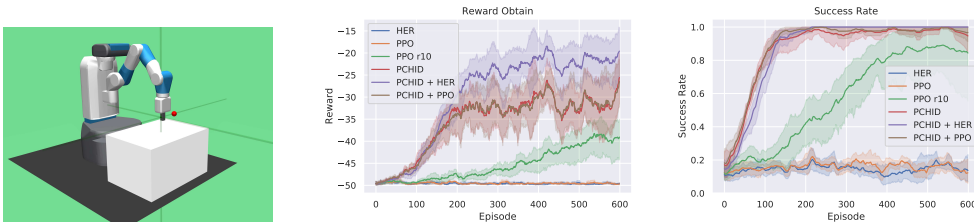

Figure 4: (a): The FetchReach environment. (b): The reward obtaining process of each method. In PPO r10 the reward of achieving the goal becomes 10 instead of 0 as default, and the reward is re-scaled to be comparable with other approaches. This is to show the sensitivity of PPO to reward value. By contrast, the performance of PCHID is unrelated to reward value. (c): The success rate of each method. Combining PPO with PCHID brings about little improvement over PCHID, but combining HER with PCHID improves the performance significantly.

by testing the learned agent on 1000 unseen maps. Our work follows VIN to use the rollout success rate as the evaluation metric.

Our empirical results are shown in Fig.3. Our method is compared with DQN, both of which are equipped with VIN as policy networks. We also apply HER to DQN but result in a little improvement. PC with 1-step HID, denoted by PCHID 1, achieves similar accuracy as DQN in much less episodes, and combining PC with 5-step HID, denoted by PCHID 5, and HER results in much more distinctive improvement.

## 4.2 OpenAI Fetch Env

In the Fetch environments, there are several tasks based on a 7-DoF Fetch robotics arm with a two-fingered parallel gripper. There are four tasks: FetchReach, FetchPush, FetchSlide and FetchPickAndPlace. In those tasks, the states include the Cartesian positions, linear velocity of the gripper, and position information as well as velocity information of an object if presented. The goal is presented as a 3-dimentional vector describing the target location of the object to be moved to. The agent will get a reward of 0 if the object is at the target location within a tolerance or $-1$ otherwise. Action is a continuous 4-dimensional vector with the first three of them controlling movement of the gripper and the last one controlling opening and closing of the gripper.

**FetchReach** Here we demonstrate PCHID in the FetchReach task. We compare PCHID with PPO and HER based on PPO. Our work is the first to extend hindsight knowledge into on-policy algorithms [3]. Fig.4 shows our results. PCHID greatly improves the learning efficiency of PPO. Although HER is not designed for on-policy algorithms, our combination of PCHID and PPO-based HER results in the best performance.

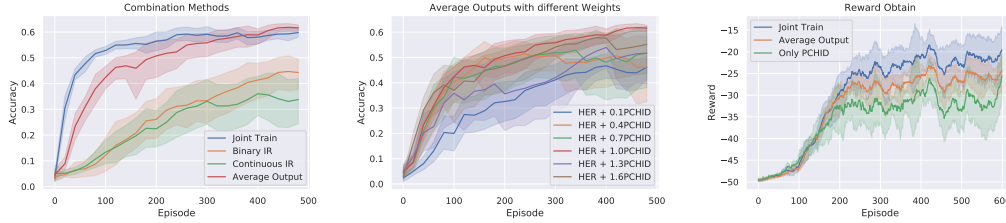

Figure 5: (a): Accuracy of GridWorld under different combination strategies. (b): Averaging outputs with different weights. (c): Obtained Reward of FetchReach under different strategies.

## 4.3 Combing PCHID with Other RL Algorithms

As PCHID only requires sufficient exploration in the environment to approximate optimal sub-policies progressively, it can be easily plugged into other RL algorithms, including both on-policy algorithms and off-policy algorithms. At this point, the PCHID module can be regarded as an extension of HER for off-policy algorithms. We put forward three combination strategies and evaluate each of them on both GridWorld and FetchReach environment.

**Joint Training** The first strategy for combining PCHID with normal RL algorithm is to adopt a shared policy between them. A shared network is trained through both temporal difference learning in RL and self-supervised learning in PCHID. The PCHID module in joint training can be viewed as a regularizer.

**Averaging Outputs** Another strategy for combination is to train two policy networks separately, with data collected in the same set of trajectories. When the action space is discrete, we can simply average the two output vectors of policy networks, e.g. the Q-value vector and the log-probability vector of PCHID. When the action space is continuous, we can then average the two predicted action vectors and perform an interpolated action. From this perspective, the RL agent here actually learns how to work based on PCHID and it parallels the key insight of ResNet [32]. If PCHID itself can solve the task perfectly, the RL agent only needs to follow the advice of PCHID. Otherwise, when it comes to complex tasks, PCHID will provide basic proposals of each decision to be made. The RL agent receives hints from those proposals thus the learning becomes easier.

**Intrinsic Reward (IR)** This approach is quite similar to the curiosity driven methods. Instead of using the inverse dynamics to define the curiosity, we use the prediction difference between PCHID module and RL agent as an intrinsic reward to motivate RL agent to act as PCHID. Maximizing the intrinsic reward helps the RL agent to avoid aimless explorations hence can speed up the learning process.

Fig.5 shows our results in GridWorld and FetchReach with different combination strategies. Joint training performs the best and it does not need hyper-parameter tuning. On the contrary, the averaging outputs requires determining the weights while the intrinsic reward requires adjusting its scale with regard to the external reward.

## 5 Conclusion

In this work we propose the Policy Continuation with Hindsight Inverse Dynamics (PCHID) to solve the goal-oriented reward sparse tasks from a new perspective. Our experiments show the PCHID is able to improve data efficiency remarkably in both discrete and continuous control tasks. Moreover, our method can be incorporated with both on-policy and off-policy RL algorithms flexibly.

**Acknowledgement:** We acknowledge discussions with Yuhang Song and Chuheng Zhang. This work was partially supported by SenseTime Group (CUHK Agreement No.7051699) and CUHK direct fund (No.4055098).

## Footnotes

[1]Code and related materials are available at `https://sites.google.com/view/neurips2019pchid`

[2]Tarmar et al. train VIN through the imitation learning (IL) with ground-truth shortest paths between start and goal positions. Although both of our approaches are based on IL, we do not need ground-truth data

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
