[Supplementary Material]

## Supplementary Materials

### The Ornstein-Uhlenbeck Process Perspective of Synchronous Improvement

For simplicity we consider 1-dimensional state-action space. An policy equipped with Gaussian noise in the action space $a \sim \mathcal{N}(\mu, \sigma^2)$ lead to a stochastic process in the state space. In the most simple case, the mapping between action space and the corresponding change in state space is an affine transformation, i.e., $\Delta s_t = s_{t+1} - s_t = \alpha a_t + \beta$. Without loss of generality, we have

$$\Delta s_t \sim \mathcal{N}(\epsilon(g - s_t), \sigma^2) \tag{10}$$

after normalization. The $\epsilon$ describes the correlations between actions and goal states. e.g., for random initialized policies, the actions are unaware of goal thus $\epsilon = 0$, and for optimal policies, the actions are goal-oriented thus $\epsilon = 1$. The learning process can be interpreted as maximizing $\epsilon$, where better policies have larger $\epsilon$. Under those notations,

$$ds_t = \epsilon(g - s_t)dt + \sigma dW_t \tag{11}$$

where $W_t$ is the Wiener Process, and the corresponding discrete time version is $\Delta s_t = \epsilon(g - s_t)\Delta t + \sigma \Delta W_t$. As Eq. (11) is exactly an Ornstein-Uhlenbeck (OU) Process, it has closed-form solutions:

$$s_t = s_0 e^{-\epsilon t} + g(1 - e^{-\epsilon t}) + \sigma \int_0^t e^{-\epsilon(t-s)} dW_s \tag{12}$$

and the expectation is

$$\mathbb{E}(s_t) - g = (s_0 - g)e^{-\epsilon t} \tag{13}$$

Intuitively, Eq. (13) shows that as $\epsilon$ increase during learning, it will take less time to reach the goal. More precisely, we are caring about the concept of First Hitting Time (FHT) of OU process, i.e., $\tau = \inf\{t > 0 : s_t = g | s_0 > g\}$ [33].

Without loss of generality, we can normalize the Eq.(11) by the transformation:

$$\tilde{t} = \epsilon t, \quad \tilde{s} = \frac{\sqrt{2\epsilon}}{\sigma}(s - g), \quad \tilde{g} = \frac{\sqrt{2\epsilon}}{\sigma}(g - g) = 0, \quad \tilde{s}_0 = \frac{\sqrt{2\epsilon}}{\sigma}(s_0 - g) \tag{14}$$

and we consider the FHT problem of

$$d\tilde{s}_t = -\tilde{s}_t d\tilde{t} + 2dW_{\tilde{t}}$$
$$\tilde{\tau} = \inf\{\tilde{t} > 0 : \tilde{s}_t = 0 | \tilde{s}_0 > 0\} \tag{15}$$

The probability density function of $\tilde{\tau}$, denoted by $p_{0,\tilde{s}_0}(\tilde{t})$ is

$$p_{0,\tilde{s}_0}(\tilde{t}) = \sqrt{\frac{2}{\pi}} \frac{\tilde{s}_0 e^{-\tilde{t}}}{(1 - e^{-2\tilde{t}})^{3/2}} \exp\left(\frac{\tilde{s}_0^2 e^{-2\tilde{t}}}{2(1 - e^{-2\tilde{t}})}\right) \tag{16}$$

and the expectation is provided as [34, 35, 36]

$$\mathbb{E}[\tilde{\tau}] = \sqrt{\frac{\pi}{2}} \int_{-\tilde{s}_0}^0 \left(1 + \text{erf}\left(\frac{\tilde{t}}{\sqrt{2}}\right)\right) \exp\left(\frac{\tilde{t}^2}{2}\right) d\tilde{t} \tag{17}$$

Accordingly, the optimization of solving goal-oriented reward sparse tasks can be viewed as minimizing the FHT of OU process. From this perspective, any action that can reduce the FHT will lead to a better policy.

Inspired by such a perspective, and to tackle the efficiency bottleneck and further improve the performance of PCHID, we extend our method to a synchronous setting based on the evolving concept of $k$-step solvability. We refer to this updated approach as Policy Evolution with Hindsight Inverse Dynamics (PEHID). PEHID start the learning of $\pi_i$ before the convergence of $\pi_{i+1}$ by merging buffers $\{\mathbb{B}_i\}_{i=1}^T$ into one single buffer $\mathbb{B}$. And when increasing the buffer with new experiences, we will test an experience that is $k$-step solvable could be reproduced within $k$ steps if we change the goal. We retain those experiences that are not reachable as containing new valuable skills for current policy to learn.

---
**Algorithm 2** PEHID Module

---
**Require**
- a policy $\pi_b(s, g)$
- a reward function $r(s, g) = 1$ if $g = m(s)$ else $0$
- a buffer for PEHID $\mathbb{B}$
- a list $\mathbb{K} = [1, 2, ..., K]$

Initialize $\pi_b(s, g)$, $\mathbb{B}$
**for** episode $= 1, M$ **do**
  generate $s_0$, $g$ by the system
  **for** $t = 0, T - 1$ **do**
    Select an action by the behavior policy $a_t = \pi_b(s_t, g)$
    Execute the action $a_t$ and get the next state $s_{t+1}$
    Store the transition $((s_t, g), a_t, (s_{t+1}, g))$ in a temporary episode buffer
  **end for**
  **for** $t = 0, T - 1$ **do**
    **for** $k \in \mathbb{K}$ **do**
      calculate additional goal according to $s_{t+k}$ by $g' = m(s_{t+k})$
      **if** TEST$(k, s_t, g')$ = True **then**
        Store $(s_t, g', a_t)$ in $\mathbb{B}$
      **end if**
    **end for**
  **end for**
  Sample a minibatch $B$ from buffer $\mathbb{B}$
  Optimize behavior policy $\pi_b(s_t, g')$ to predict $a_t$ by supervised learning
**end for**

---

Table 1: The successful rate of different methods in the FetchPush, FetchSlide and FetchPickAndPlace environments (trained for 1.25M timesteps)

| Method | FetchPush | FetchSlide | FetchPickAndPlace |
|---|---|---|---|
| PPO | 0.00 | 0.00 | 0.00 |
| DDPG | 0.08 | 0.03 | 0.05 |
| DDPG + HER | **1.00** | 0.30 | 0.60 |
| PEHID | 0.95 | **0.38** | **0.75** |

**Empirical Results**

We evaluate PEHID in the FetchPush, FetchSlide and FetchPickAndPlace tasks. To demonstrate the high learning efficiency of PEHID, we compare the success rate of different method after 1.25M timesteps, which is amount to 13 epochs in the work of Plappert et. al [3]. Table 1 shows our results.