[Reviews · NeurIPS 2019]

Reviewer 1



QUALITY I like the approach but I have one fundamental problem in understanding the method. The policy is a mapping from S x G -> a while the inverse dynamics map from (S x G) x (S x G) -> a. How are pi and theta equal and how are they parametrized? In the classical sense, when learning the inverse dynamics, this problem is more about the environment and not the policy, if I am not mistaken. If so, why is it important to relabel data to learn them? Also the combination of PCHID with PPO is not fully sound (as is noted in the paper) but could be solved by just exchanging PPO with an off-policy algorithm, for example TD3. Nonetheless, the experimental results section shows some interesting results with much improved performance and sample-complexity over standard DQN, HER and PPO. CLARITY The paper is well written and structured, only some sentences need some proof reading for some expressions. Regarding my lack of understanding the difference between a policy and inverse dynamics, maybe some more words are necessary to better link these two functions. In the grid world setting, k-step solvability is straightforward to understand and implement while in continuous action spaces it is not so straightforward. Can you comment a bit more on how to determine k-step solvability in continuous action domains? ORIGINALITY The paper introduces some fundamental concepts and combines them into a novel algorithm type. SIGNIFICANCE Improving sample-complexity is of high importance to the field and the results look promising. ##################### Post rebuttal The authors clarified my main question on the connection between inverse dynamics and the policy. The other questions were also answered appropriately. Therefore I would keep my current rating and increase my confidence score.

Reviewer 2



The proposed method extending hindsight experience replay with k-step inverse dynamics learning is original and significant. While there is a limitation that this paper assumes a deterministic environment and a goal space that is a part of the state space and on which one can define a test function, there exist many challenging scenarios that satisfy these assumptions. For example, this paper and the original hindsight experience replay made a similar assumption but, it showed its potential for challenging continuous control tasks. The paper is clearly written and the intuition is easy to understand. I have minor questions about the experimental setting, especially about the choice of maximum K. In theory, the maximum K should correspond to the maximum number of steps required to reach the goal, but the experiment in the grid world used maximum K=5 and the maximum K is not specified in the OpenAI Fetcher environment. Since choosing the maximum K seems important for this algorithm, clarifying the used maximum K and explaining how small K still provides improvement might be necessary. * After author response I increased my rating after the author response. The only concern I mentioned in the review was about the effect of K in different scenarios. The author response effectively addressed this concern by 1) providing an intuitive explanation on how a model with small K can still improve performance, and 2) showing an additional experiment controlling K in a continuous control environment. I believe the explanation and experimental result would be useful to understand an important characteristic of the algorithm: the effect of K. Therefore, I would recommend having the results in the final version.

Reviewer 3



Overall, the paper is clearly written. The theory and methods are well developed. And the results are discussed in details with ablation study. However, I do have a question about the significance of the approach. If I understand correctly, the proposed method, policy-continuation + hindsight inverse dynamics (PCHID) is a continuous version of dynamic programming. The difference is that the table is replaced with a function approximator so PCHID can handle continuous state-action space. Have the authors tried to solve the GridWorld task using simple dynamic programming as a comparison? I also have a concern regarding the algorithm part. The proposed algorithm validates whether a state is k-step reachable using a TEST algorithm. Because a function approximator has been used, we cannot guarantee that the given state is not k-step reachable even if the function approximator yields no solution. I think additional theoretical analysis is needed to address this issue. Update: The authors have replied and addressed my main comments: the DP perspective of the PCHID, and the false negatives in the TEST routine.

[Author Response · NeurIPS 2019]

We thank all reviewers for their insightful comments. Please see the responses below.

**To Review 1:**

**Q1: The connection between the policy and the Hindsight Inverse Dynamics(HID).** Instead of mapping $(s_1, g_1) \times$
$(s_2, g_2) \to a$ in vanilla inverse dynamics, the HID encodes $(s_2, g_2)$ to replace $g_1$, i.e. $g_1' = m(s_2)$. So that both HID
and the policy maps $(S \times G) \to a$. (e.g. $(s_1, g_1') \to a$ for an 1-step example). $g_2$ is omitted as the original goals are not
concerned in HID.

**Q2: Why is it important to relabel data to learn HID?** First, the relabel process enables the HID and policy to be
parameterized in the same form, as depicted in **Q1**. Second, with such a relabeling process, multistep HID can be
introduced, on the contrary the vanilla ID can only deal with adjacent transitions.

**Q3: The combination of PCHID with PPO.** Yes, in principle the PCHID should work better when it is combined with
off-policy algorithms where the trained samples are collected with off-policy hindsight experiences. The encouraging
result of combing PCHID with PPO in our paper illustrates that PCHID is a promising approach to introduce hindsight
experience knowledge into the prevailing on-policy algorithms like PPO, enabling on-policy algorithms to learn from
failures (resolved a challenge proposed in [3]). As PCHID needs the policy network to fit HID, such procedure can
narrow the gap between the on-policy experiences generated by the policy and the HID.

**Q4: k-step solvability in continuous action domains.** Ideally the k-step solvability means the number of steps it
should take from $s$ to $g$, given the maximum permitted action value. In practice the k-step solvability is a evolving
concept that can gradually change during the learning process, thus is defined as "whether it can be solve with $\pi_{k-1}$
within k steps after the convergence of $\pi_{k-1}$ trained on (k-1)-step HIDs".

**To Review 2:**

**Q1: The choice of maximum K.** We attribute the success of 1-step HID to the flatness of the state space, and under
this circumstance extrapolation of 1-step policy works well in multistep situations. Fig.1(b) in the paper shows an
analogy of how such a flatness benefits extrapolation: an agent in a grid map is asked to reach the goal $g_3$ starting from
$s_0$: if the agent has already known how to reach $s_1$ in the east, intuitively, it is not difficult for it to extrapolate the policy
to reach $g_3$ in the farther east. On the other hand, when the goal is at $g_1$, the barrier makes the extrapolation of 1-step
policy pointing to the north fails to reach the goal. And multistep HIDs help such extrapolations in non-trivial cases. In
principle, with larger K, the successful rate of extrapolation will increase. In the GridWorld environment, our ablation
study (Fig.1(a) below) shows $K = 4$ is able to achieve good performance. And Fig.1(b) below shows similar results in
the FetchPush environment.

**To Review 3:**

**Q1: Dynamic Programming(DP) perspective of PCHID.** PCHID can be formulated as a solver from the perspective
of DP. For most goal-oriented tasks, the learning objective is to find a policy to reach the goal as soon as possible. In
such circumstances, $L^\pi(s_t, g) = L^\pi(s_{t+1}, g) + 1$, where $L^\pi(s, g)$ is defined as the number of steps needed from $s$ to $g$
with policy $\pi$ and 1 is the additional step. PC can be interpreted as a learning procedure for a solver and HIDs are the
corresponding data generated to train the solver. We will detail the DP perspective in the revision. As VIN can be seen
as a neural-network approximation of dynamic programming(VI) [30]. The DQN based on VIN can be regarded as a
simple DP baseline (Fig.3(a) in the paper).

**Q2: Discussion about the False Negatives in the TEST process.** In principle, if sufficient exploration and the
robustness of neural network approximators are guaranteed, we will have an optimal (k-1)-step sub policy before we
use it to TEST on k-step transitions, hence false negatives do not exist. In practice, false negatives do provide some
useful information on what the agent has not yet mastered. Although the agent is not guaranteed to learn optimal policy
from HIDs with false negatives, it can still learn to find a feasible path. In sparse reward settings, learning a feasible
policy is crucial and several previous work like learning from demonstrations [4], self-imitation learning [6] can be
absorbed for further improvement.

**Q3: Order of article arrangement.** Thank you for the advice. We will rearrange Sec.4.3 in the revision.

**Q4: More diverse experiments.** We evaluate PCHID on the FetchPush environment. The results are shown in Fig.1(b)
below. We also test on the choice of maximum K. The multistep PCHID performs much better than 1-step PCHID as
the FetchPush is a multistep task, i.e. an agent needs to first move the gripper to the block and then push it to the target
position. We will evaluate PCHID on more benchmarks in the future work.

Figure 1: (a): on the selection of maximum K. The curve shows averaged results in 5 experiments with different random seeds. (b): experiments on FetchPush environment. The curve shows averaged results in 3 experiments with different random seeds

[Meta-Review · NeurIPS 2019]

The paper presents a new approach for inverse dynamics learning which is extended to goal conditioned, multi-step inverse dynamics. The approach is combined with standard RL algorithms to solve multi-goal tasks such as the OpenAI Fetch environment. All reviewers liked the ideas presented in the paper and appreciated the contributions. The experiments were also well executed and the results are convincing. I am also convinced that the paper offers interesting aspects in the field of multi-goal RL and recommend this paper for a spotlight presentation.